# Electrocochleography in Cochlear Implant Users with Residual Acoustic Hearing: A Systematic Review

**DOI:** 10.3390/ijerph17197043

**Published:** 2020-09-26

**Authors:** Jeong-Seo Kim

**Affiliations:** Department of Communication Sciences and Disorders, University of Iowa, Iowa City, IA 52242, USA; jeong-seo-kim@uiowa.edu; Tel.: +1-913-433-6161

**Keywords:** electrocochleography, ECoG, cochlear implant, residual hearing, Hybrid CI

## Abstract

(1) Objectives: This study reviews the use of electrocochleography (ECoG) as a tool for assessing the response of the peripheral auditory system and monitoring hearing preservation in the growing population of cochlear implant (CI) users with preserved hearing in the implanted ear. (2) Methods: A search was conducted in PubMed and CINAHL databases up to August 2020 to locate articles related to the ECoG measured during or after the cochlear implant (CI) surgery for monitoring purposes. Non-English articles, animal studies, literature reviews and editorials, case reports, and conference papers were excluded. The quality of studies was evaluated using the National Institute of Health (NIH) “Study Quality Assessment Tool for Case Series Studies”. (3) Results: A total 30 articles were included for the systematic review. A total of 21 articles were intraoperative ECoG studies, while seven articles were postoperative studies. Two studies were conducted ECoG both during and after the surgery. Intraoperative ECoG studies focused on monitoring changes in ECoG response amplitudes during and/or after electrode insertion and predicting the scalar location of the electrode array. Postoperative ECoG studies focused on using the ECoG measurements to estimate behavioral audiometric thresholds and monitor pathophysiological changes related to delayed onset hearing loss postimplant. (4) Conclusions: ECoG is feasible to provide real-time feedback intraoperatively and has a potential clinical value to monitor the status of hearing preservation postoperatively in this CI population with residual acoustic hearing.

## 1. Introduction

Cochlear implant (CI) has been a successful (re)habilitation option to restore hearing sensitivity for people with severe to profound sensorineural hearing loss over recent decades [1]. Recently, CI candidacy criteria have been significantly relaxed to individuals with good low-frequency hearing but substantial bilateral, high-frequency hearing loss [2,3,4]. These individuals with severe to profound high-frequency hearing loss often do not receive sufficient benefits via hearing aids [5]. For this population, combined electroacoustic stimulation (EAS) appears to be a viable option compared to conventional cochlear implantation that may result in the complete loss of the residual hearing. Less traumatic CI electrode array design and the use of “soft surgery” techniques allow for the preservation of residual low-frequency acoustic hearing. EAS, also called “Hybrid” CI, is designed to provide both electric and acoustic stimulation to the same ear. The electrode array provides high frequency electric stimulation, while an integrated hearing aid provides amplification for low frequency sounds. Preserving acoustic hearing provides benefits including improved speech performance in background noise and music perception when listeners use a combined acoustic and electric stimulation compared to those with an electric stimulation alone [2,3,4,6,7].

These individuals are more likely to have preserved structures including surviving hair cells and auditory nerve fibers near the apex of the cochlea after cochlear implantation. This results in interest in the role that the auditory periphery may play in driving outcomes with a CI. For CI users with residual acoustic hearing, responses from the cochlea to acoustic stimuli can be recorded using the electrocochleography (ECoG) technique. Electrocochleography (ECoG) is a technique which has been used for decades to record responses from the cochlear hair cells and the auditory nerve [8]. The recording obtained using ECoG is a composite response that includes contributions from hair cells (i.e., the cochlear microphonic (CM), summating potential (SP)) and the auditory nerve (i.e., the compound action potential (CAP), auditory nerve neurophonic (ANN)).

The cochlear microphonic (CM) is a recording of current flow through the mechanoelectric transducer channels in the stereocilia of hair cells [9]. The CM likely includes contributions from both outer and inner hair cells, but it is dominated by the responses from the outer hair cells [10]. The summating potential (SP) is recorded as a direct current (DC) shift in the baseline of the evoked response. It is thought to reflect receptor potentials that arise from multiple generators within the cochlea and most likely reflects mixed contributions from both inner and outer hair cells [11]. Two components reflect activity generated by the auditory nerve: the compound action potential (CAP) and the auditory nerve neurophonic (ANN). The CAP is a recording of the synchronized response of multiple auditory nerve fibers and is evident both at the onset and offset of the stimulus [12]. The ANN is the phase-locked activity of the auditory nerve fibers in response to an ongoing, low-frequency sinusoidal stimulus [13]. This complex of responses, collectively called the ECoG, provides a rich source of information about the survival of functional cochlear elements and may help understand the underlying pathophysiology related to the peripheral auditory system. 

The ECoG has been used in clinical settings to support the diagnosis and assessment of Meniere’s disease, to enhance the wave I of the auditory brainstem response (ABR), to assist with the diagnosis of auditory neuropathy spectrum disorders (ANSD), and to monitor cochlear and auditory nerve function during surgery [14,15,16]. Over the last 30 years, ECoG has been applied as an excellent tool for the diagnosis of patients with Meniere’s disease/endolympathic hydrops [8,17]. ECoG recordings from patients with Meniere’s disease are characterized by an enlarged SP and SP/Action potential (AP) ratio compared to normal hearing ears due to an increase in endolymphatic volume that creates mechanical biasing of vibration of the organ of Corti and amplifies the SP [14,15]. An interest in ECoG as a diagnostic tool has increased again since 2010 due to current attention on potential usefulness of ECoG in CI users with preserved hearing for intraoperative and postoperative monitoring purposes [8]. 

Once hearing preservation became possible for CI users with the development of thinner, flexible electrode arrays and the use of the soft surgical techniques, there were surging interests in using ECoG to measure cochlear function in this expanded CI population. The ECoG has been increasingly used to monitor the status of hearing to mitigate possible intracochlear damage while inserting an electrode and optimize electrode placement during cochlear implant surgery [18,19,20,21,22,23,24,25]. Recently, noninvasive recording methods have been developed that allow ECoG responses to be measured from an intracochlear electrode [26,27,28,29,30]. This was accomplished via the reverse neural telemetry capabilities of the CI [26,28]. The ECoG was also clinically applied to estimate and monitor hearing preservation postimplant and understand the considerable variance in postoperative CI outcomes [26,28,29,30,31]. This study aims to systemically analyze published literature on the use of ECoG as a means of assessing the response of the peripheral auditory system during and after the cochlear implant surgery in CI recipients with residual acoustic hearing. 

## 2. Materials and Methods 

This systematic review was guided by the Preferred Reporting Items for Systematic Reviews and Meta-Analyses (PRISMA) guidelines [31]. 

### 2.1. Search Strategy and Eligibility Criteria

A search was conducted in search databases including the National Library of Medicine (PubMed) and CINAHL (Cumulative Index to Nursing and Allied Health Literature) in order to locate studies reporting ECoG responses recorded from cochlear implant users with preserved hearing. Key terms of “Electrocochleography” AND (“Cochlear Implant” OR “Cochlear Implantation”) AND “Residual hearing” AND “Acoustic stimulation” were used to search articles. Articles published before August 2020 were collected via PubMed and CINAHL. The reference lists of retrieved articles and personal communication were also used to search potentially relevant articles. The search strategy is shown in Figure 1.

The main inclusion criteria were articles written in English, cohort, or case studies investigating the use of ECoG in CI users for intraoperative and postoperative monitoring purposes. Animal studies, articles written in other languages, literature reviews and editorials, studies with preliminary results with subjects less than five, case studies, and conference papers were excluded. Papers focused on other electrophysiology measures on CI users or ECoG studies focused on purposes other than monitoring for hearing preservation were not included in this review. 

### 2.2. Data Extraction and Quality Assessment 

Qualitative data from each article were extracted and organized in the table. Extracted information included (1) author, (2) title, (3) year of publication, (4) country, (5) study subjects, (6) time at ECoG measurement, (7) ECoG method, (8) main results, and (9) study quality scores. The quality of studies was evaluated by applying the 9 items of the National Institute of Health (NIH) “Study Quality Assessment Tool for Case Series Studies” [32]. The summary score of each study was calculated as a percentage with a range of 0–100%. These were grouped into four categories: poor (0–25%), fair (26–50%), good (51–75%), or excellent (76–100%). Studies were included in this systematic review regardless of their quality to mitigate the risk of bias. 

## 3. Results

After removing duplicated findings and screening, 58 full-text articles were retrieved, and 30 articles met the inclusion and exclusion criteria (Figure 1). Twelve articles were excluded with following reasons (Figure 1): 3 animal studies, 3 articles written with non-English languages, 3 reviews or editorials, 2 articles with preliminary results, and 1 conference paper. In addition, sixteen articles were not included with following reasons: two articles focused on other electrophysiology testings on CI users (i.e., auditory steady-state response (ASSR), vestibular evoked myogenic potential (VEMP)), thirteen studies used ECoG for other purposes (i.e., psychophysical testing, differential diagnosis, genetic testing) that are not directly related to our current interest of this review, and one case study. All included studies are summarized in Table 1. Studies are organized by an alphabetical order of authors. Meta-analyses were not possible due to heterogeneity of ECoG recording techniques and procedures, CI device types, and residual hearing of participants across studies. 

Years of publication range between 2012 and 2020. Included studies were grouped into two main categories according to the time at ECoG measurement: intraoperative and postoperative. Considering variabilities of ECoG procedures, a recording method was also characterized as the extratympanic procedure, round window approach, and intracochlear technique. ECoG responses can be measured by using a gold foil wrapped or cotton wick electrode placed at extratympanic sites, or from a monopolar needle electrode placed around or at the round window during cochlear implantation. Additionally, an intracochlear electrode can be used to record ECoG responses noninvasively via the reverse telemetry capabilities of the CI. Out of the 30 total studies reviewed, ECoG was implemented during cochlear implantation in 70% (21/30), while ECoG was conducted in follow-up after the surgery in 23% (7/30) of studies. Two studies (7%) conducted ECoG both during and after the surgery (Figure 2).

In terms of ECoG recording procedures, the intracochlear method was used in 50% (15/30) of studies, and the round window technique was used in 37% (11/30). Both round window and intracochlear ECoG methods were conducted in 10% (3/30), while extratympanic and intracochlear ECoG methods were implemented in the other 3% (1/30) of studies (Figure 3). In the 30 papers reviewed, most of the study population was adults (63%, 19/30). Four studies reported ECoG measured in the pediatric population (10%, 3/30), and the other three articles collected ECoG responses from participants of all ages (27%, 8/30). After quality assessment, all studies reviewed were classified as excellent (90%, 27/30) or good (10%, 3/30). This indicates that included studies are mostly high quality. Lower ratings were due to the small number of participants, less informative descriptions of subject demographics, and statistical analysis.

## 4. Discussion

### 4.1. Intraoperative ECoG in CI Users

Once hearing preservation became possible for CI users with the development of less traumatic electrode arrays and the use of the soft surgical techniques, the feasibility of ECoG to measure cochlear function was investigated in this growing population of CI recipients. The ECoG technique has been increasingly used to monitor the status of hearing during cochlear implant surgery [18,19,20,35,36,37,38,39,40,41,42,47,48]. In most of these intraoperative monitoring studies, the major focus was monitoring the CM. The CM was selected because it is larger in amplitude than the ANN and can be recorded in the majority of CI users which makes an interpretation much easier. Additionally, the CM is generated by the hair cells. These cochlear hair cells are also likely to be the site of damage in the cochlea if insertion trauma occurs during cochlear implantation.

ECoG responses were recorded from a monopolar needle electrode placed at the round window before or after an insertion of the CI electrode array into the scala tympani to record changes in response during or after electrode insertion [18,19,20,21,22,35,36,37,38,39,40,44,46]. In general, ECoG responses after insertion were significantly smaller than the preinsertion response by 3 to 5 dB [18,28,40]. Mixed results were reported for the relationship between changes in ECoG responses before and after an electrode insertion and postoperative residual hearing. Average 4 dB changes in ECoG response magnitude during the surgery were not correlated with changes in audiometric thresholds after implantation [18]. However, subjects with detectable decrease in or loss of ECoG responses at low and high frequencies immediately after electrode insertion showed a significantly greater hearing loss four weeks after the surgery compared to subjects without decrease in or loss of ECoG signals pre- and postinsertion [20,36].

A recent innovative technique to measure ECoG used an electrode located inside the cochlea that allows noninvasive recordings. Clearly, proximity to the generating source could be a significant advantage. This was accomplished by using the reverse telemetry capabilities of the CI [26,28,29]. This bidirectional telemetry system is available for all three major CI devices, as referred to as the Neural Response Telemetry (NRT, Cochlear Ltd.), Neural Response Imaging (NRI, Advanced Bionics), and Auditory Nerve Response Telemetry (ART, MED-EL). The intracochlear ECoG recording via the reverse telemetry system was feasible to offer the real-time feedback of cochlear responses during electrode insertion [23,24,25,27,41,42,47,48]. The prognostic value of intracochlear ECoG obtained in the operating room to assess insertion trauma and predict early postoperative hearing preservation is yet unclear. Subjects who showed stable CM amplitudes during an electrode insertion tended to have preserved hearing afterwards [46,47]. Subjects with preserved CM at the end of insertion had on average 15 dB better low-frequency acoustic hearing compared to subjects with transient or permanent reduction in the CM amplitudes during the insertion process [28]. However, in subjects who had an average 3 dB drop in CM amplitude during electrode insertion, neither CM amplitude drop from the beginning of insertion to peak amplitude nor drop from peak amplitude to the end of insertion were significantly correlated with postoperative audiometric threshold shift at low frequencies (125–500 Hz) [23].

Intraoperative ECoG recordings have the potential to provide information about scalar location of the electrode array during and after electrode insertion. Frequent instances of translocation (22–38%) from scala tympani (ST) to scala vestibuli (SV) during electrode insertion and its negative impact on speech outcomes were reported [23,24]. Subjects who had electrode arrays translocated from ST into SV showed a sizeable decrease in CM amplitudes during insertion without recovery, while this CM amplitude change was not significantly different from responses from the nontranslocation group [24,25]. However, the difference between intraoperative CM thresholds and postoperative audiometric thresholds for patients with electrode crossed from ST to SV was significantly lower, which suggests that scalar translocation from ST to SV was associated with significantly higher shifts in low-frequency audiometric thresholds when compared to electrodes inserted entirely within ST [23]. In addition, a trend of smaller phase changes was observed in the translocation group compared to a larger phase shift in the nontranslocation group [24]. Therefore, incorporating phase change information during electrode insertion may enhance specificity and sensitivity when predicting scalar translocation based on intraoperative ECoG measurements.

The prognostic value of intraoperative ECoG to predict postoperative speech outcomes was investigated in a series of studies from the same research group [21,22,38,49]. The ECoG total response (TR) metric—the sum of first and second harmonics of the Fast Fourier transform (FFT) analysis—was used in these studies. Moderate correlations between TR magnitudes and CNC / PB-K word scores were observed both in adults and children [21,22,38]. TR accounted for 46% of variability in speech recognition for adults, while 15–36% of variability was accounted for by children. When CAP responses that reflect purely neural contributions were used as a metric, a weak correlation (*r* = 0.44) was reported between the CAP amplitudes and postoperative CNC word score [49].

### 4.2. Postoperative ECoG in CI Users

ECoG recordings have been used postoperatively to monitor the status of hearing preservation in CI users with residual acoustic hearing [23,26,28,29,30]. In most studies, CM and ANN responses were well identified from the majority of subjects. CM and ANN thresholds were found to be strongly correlated with postoperative behavioral thresholds [23,26,29,30]. Objective measurements of audiometric thresholds based on ECoG measurements may be clinically applicable to estimate behavioral audiometric thresholds from younger children or hard-to-test populations with limited cooperation.

Postoperative ECoG has also been explored as a tool to monitor underlying pathophysiological changes associated with postimplant loss of acoustic hearing that is experienced by a sizeable proportion of CI users [26,30]. Despite efforts to preserve cochlear structures and acoustic hearing by using soft surgical techniques and carefully designed electrodes, those implanted with hearing preservation electrode arrays often experience 10–15 dB of acoustic hearing decrease in the implanted ear immediately after the surgery [2,6]. In a small population of CI users with residual acoustic hearing, delayed onset hearing loss occurs within the first year of CI use. A total of 20% of CI users who were implanted with a Cochlear Nucleus Hybrid S8 electrode experienced an average 24 dB of hearing loss several months after surgery in addition to hearing loss documented at initial activation of CI [50]. A total of 38% of adult Nucleus Hybrid CI users were presented with delayed, progressive hearing loss of various degrees and rates, though approximately 80% of subjects still retained useful residual hearing suitable for an acoustic component in the implanted ear [51]. For these individuals with delayed onset hearing loss, significantly elevated CM and ANN thresholds have been reported as their acoustic hearing changed over time [26,30]. Even subjects who had experienced less significant hearing loss (<10 dB change in pure tone average at 250–1000 Hz) showed notable increases in CM and ANN thresholds. ECoG recording was sensitive enough to reflect the pattern of changes in the residual acoustic hearing. This indicates its potential clinical value to monitor changes in the status of the peripheral auditory system postimplant.

## 5. Conclusions

The ECoG has been widely applied as a clinical tool that provides rich information of the auditory periphery including cochlear hair cells and the auditory nerve. There are many circumstances where it might be helpful if we could use ECoG components (the CM and the ANN) to identify the site of lesion (presynaptic vs. postsynaptic), assist the differential diagnosis, and understand considerable variance in the postoperative performance of CI users. With growing efforts for innovative electrode design to preserve cochlear structures and less traumatic CI surgery, ECoG appears to be useful for CI patients in terms of hearing preservation. ECoG is proven to be feasible to provide real-time feedback intraoperatively and monitor the status of hearing preservation postoperatively. This review shows how this long-standing diagnostic tool has been successfully applied to this CI population with preserved acoustic hearing. Further studies are required to make this technique more clinically accessible and understand how ECoG responses can be attributed to considerable variabilities in CI outcomes.

## Figures and Tables

**Figure 1 ijerph-17-07043-f001:**
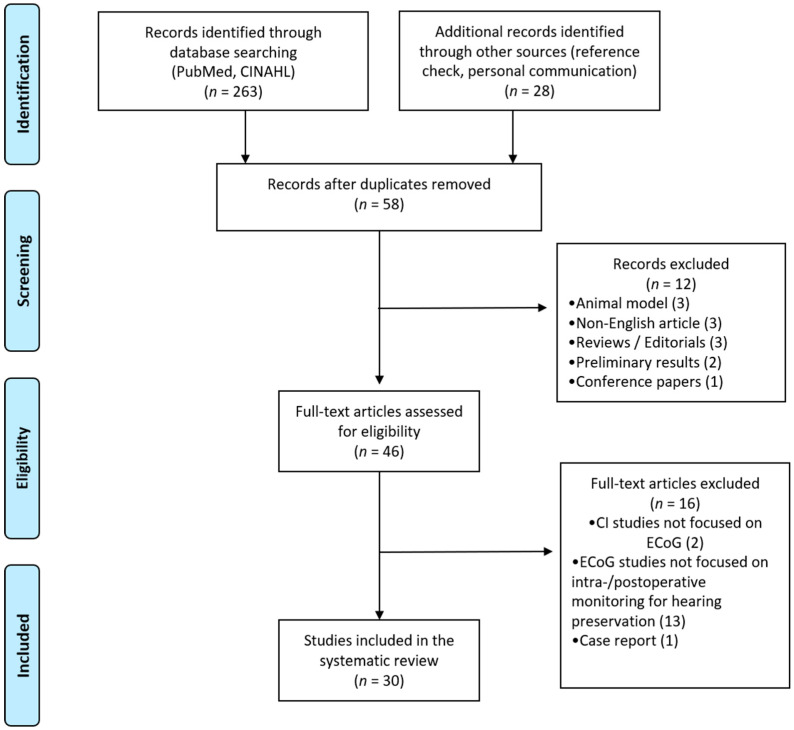
PRISMA (Preferred Reporting Items for Systematic Reviews and Meta-Analysis) flow diagram for study selection.

**Figure 2 ijerph-17-07043-f002:**
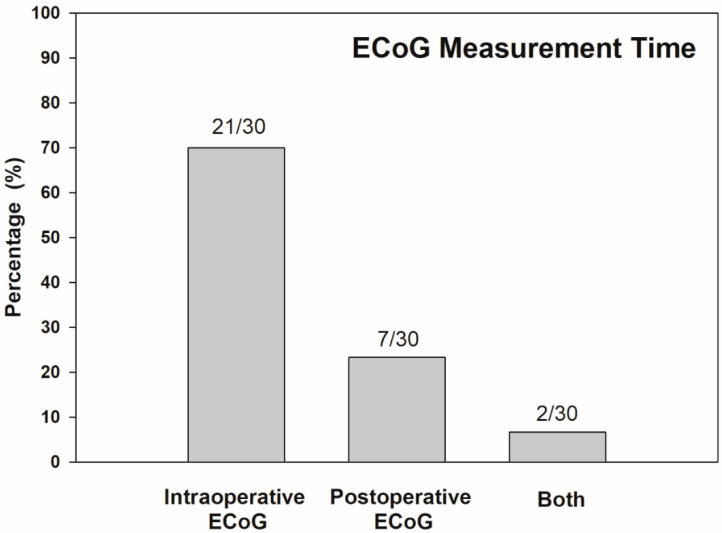
Distribution of time at ECoG measurement across all papers reviewed (*n* = 30). Numbers above bars indicate number of studies corresponding each measurement time.

**Figure 3 ijerph-17-07043-f003:**
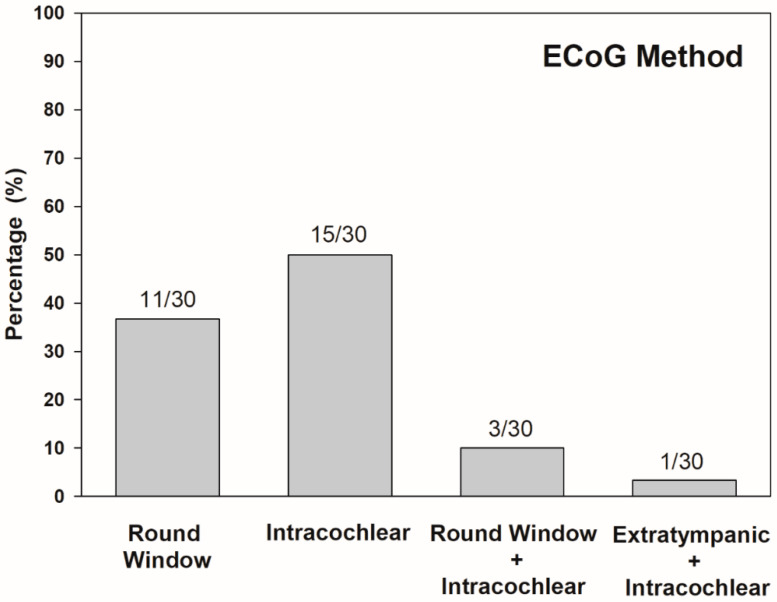
Distribution of ECoG method in percentage across all papers reviewed (*n* = 30). Numbers above bars indicate number of studies corresponding each method.

**Table 1 ijerph-17-07043-t001:** Summary of 30 included articles.

No	Authors	Title	Year	Country	Subjects	Time at ECoG Measurement	ECoG Method	Main Results	NIH Quality Score
1	Abbas et al. [26]	Using neural response telemetry to monitor physiological responses to acoustic stimulation in Hybrid cochlear implant users	2017	USA	44 Adults	Postoperative	Intracochlear	CM and ANN thresholds well correlated with 500 Hz audiometric thresholds. CM and ANN magnitude reduced as residual hearing changed over time.	89% (Excellent)
2	Adunka et al. [18]	Round widow electrocochleography before and after cochlear implant electrode insertion	2015	USA	14 Children 17 Adults	Intraoperative	Round Window	Average 4 dB change in ECoG Total Response (TR) magnitude before and after electrode insertion. No correlation between change in ECoG response and change in audiometric thresholds (22 dB on average).	89% (Excellent)
3	Attias et al. [33]	Postoperative intracochlear electrocochleography in pediatric cochlear implant recipients: association with audiometric thresholds and auditory performance	2020	Israel	60 Children	Postoperative	Intracochlear	CM responses identified in 29 ears out of 88 ears. CM amplitudes were highly correlated (*r* = 0.7–0.83) with the audiometric thresholds at 125–2000 Hz.	89% (Excellent)
4	Bester et al. [27]	Characterizing electrocochleography in cochlear implant recipients with residual hearing low-frequency hearing	2017	Australia	45 Adults	Intraoperative	Intracochlear	The most prevalent ECoG pattern with a peak in the CM amplitude near the most apical electrode was found in 21 subjects. CM amplitudes were well correlated with low frequency residual hearing.	79% (Excellent)
5	Campbell et al. [28]	Cochlear response telemetry: Intracochlear electrocochleography via cochlear implant neural response telemetry pilot study results	2014	Australia	5 Adults	Postoperative	Intracochlear	CM, ANN, CAP, SP responses via an intracochlear recording method were reliably measured in all subjects. Apical electrodes yielded the most reliable recordings.	67% (Excellent)
6	Choudbury et al. [19]	Intraoperative round window recordings to acoustic stimuli from cochlear implant patients	2012	USA	11 Children 14 Adults	Intraoperative	Round Window	CM and ANN responses were measurable in 23 out of 25 subjects. Response magnitudes were largest at low frequencies.	78% (Excellent)
7	Coulthurst et al. [34]	Comparison of pure-tone thresholds and cochlear microphonics thresholds in pediatric cochlear implant patients	2020	USA	13 Children	Postoperative	Intracochlear	Significant correlation (*r* = 0.77) presented between CM thresholds and postoperative audiometric thresholds at 125–2000 Hz.	78% (Excellent)
8	Dalbert et al. [35]	Correlation of electrophysiological properties and hearing preservation in cochlear implant patients	2015	Switzerland	19 Adults	Intraoperative	Round Window	CM and ANN responses were measurable in 18 out of 19 subjects. A total of 17 subjects had no considerable changes in ECoG recordings at low frequencies after electrode insertion.	78% (Excellent)
9	Dalbert et al. [20]	Assessment of cochlear trauma during cochlear implantation using electrocochleography and cone beam computed tomography	2016	Switzerland	14 Adults	Intraoperative	Round Window	A total of 4 subjects with hearing loss >11 dB or complete hearing loss showed a decrease in intraoperative high- or low-frequency ECoG responses,	78% (Excellent)
10	Dalbert et al. [36]	Assessment of cochlear function during cochlear implantation by extra- and intracochlear electrocochleography	2018	Switzerland	22 Adults	Intraoperative	Round Window, Intracochlear	Changes in round window ECoG recordings correlated with postoperative hearing change. Subjects who had hearing loss of 22 dB after surgery showed a detectable decrease or loss of ECoG responses after electrode insertion.	78% (Excellent)
11	Dalbert et al. [37]	Simultaneous intra- and extracochlear electrocochleography during electrode insertion	2020	Switzerland	12 Adults	Intraoperative	Round Window, Intracochlear	The mean amplitude difference between intra- and extracochlear ECoG responses was 14 dB. Weak to moderate correlation between maximum amplitudes and residual postoperative hearing existed.	89% (Excellent)
12	Fitzpatrick et al. [21]	Round window electrocochleography just before cochlear implantation: relationship to word recognition outcomes in adults	2014	USA	52 Children 32 Adults	Intraoperative	Round Window	ECoG Total Response (TR) was recorded in 80 out of 84 subjects. Correlation between the ECoG magnitude and CNC word score accounted for 37% of the variance.	78% (Excellent)
13	Fontenot et al. [38]	Residual cochlear function in adults and children receiving cochlear implants: correlations with speech perception outcomes	2019	USA	94 Children 84 Adults	Intraoperative	Round Window	ECoG TR accounted for 46% of variability in speech perception outcome	89% (Excellent)
14	Formeister et al. [22]	Intraoperative round window electrocochleography and speech perception outcomes in pediatric cochlear implant recipients	2015	USA	77 Children	Intraoperative	Round Window	ECoG TR was significantly correlated with PB-K speech perception scores and accounted for 32% of the variance. ECoG TR was weekly correlated with preoperative audiometric thresholds.	89% (Excellent)
15	Giardina et al. [39]	Response changes during insertion of a cochlear implant using extracochlear electrocochleography	2018	USA	18 Children 45 Adults	Intraoperative	Round Window	ECoG changes <5 dB during electrode insertion were shown in 38 out of 63 subjects. A total of 12 subjects showed ECoG changes >5 dB had no response recovery, while 13 subjects showed partial or complete response recovery at the end of insertion.	78% (Excellent)
16	Giardina et al. [40]	Intracochlear electrocochleography: response patterns during cochlear implantation and hearing preservation	2019	USA	31 Children 5 Adults	Intraoperative	Round Window, Intracochlear	ECoG response magnitude patterns (increase or decrease >5 dB, stable) varied by device types.	78% (Excellent)
17	Harris et al. [41]	Real-time intracochlear electrocochleography obtained directly through a cochlear implant	2017	USA	14 Adults	Intraoperative	Intracochlear	ECoG responses were measurable in all participants. CM and ANN amplitudes steadily increased during electrode insertion.	67% (Good)
18	Harris et al. [42]	Patterns seen during electrode insertion using intracochlear electrocochleography obtained directly through a cochlear implant	2017	USA	5 Children 12 Adults	Intraoperative	Intracochlear	Three ECoG patterns observed (overall CM increase during insertion (52%), maximum amplitude at the beginning (11%), maximum amplitude mid insertion (35%))	67% (Good)
19	Haumann et al. [43]	Monitoring of the inner ear function during and after cochlear implant insertion using electrocochleography	2019	Germany	10 Adults	Intraoperative Postoperative	Extratympanic Intracochlear	Intraoperative ECoG amplitudes were larger than extratympanic ECoG. Weak correlation observed between intraoperative response and postop audiometric thresholds.	78% (Excellent)
20	Hoesli et al. [44]	Electrocochleographic responses before and after short-term suprathreshold electrical stimulation in human cochlear implant recipients	2018	Switzerland	14 Adults	Intraoperative	Round Window	After electrode insertion, intraoperative ECoG response remained unchanged before and after suprathreshold electrical stimulation in most of subjects.	78% (Excellent)
21	Kim et al. [30]	Postoperative electrocochleography from Hybrid cochlear implant users: An alternative analysis procedure	2018	USA	34 Adults	Postoperative	Intracochlear	Significant correlation between CM and ANN thresholds and postop audiometric thresholds. ECoG thresholds increased when delayed hearing loss occurred postop.	78% (Excellent)
22	Kim et al. [45]	Intracochlear recordings of acoustically and electrically evoked potentials in Nucleus Hybrid L24 cochlear implant users and their relationship to speech perception	2017	USA	25 Adults	Postoperative	Intracochlear	CM and ANN amplitudes were not correlated with CNC or AzBio scores. ECoG amplitude was significantly correlated with a metric of acoustic gain in noise relative to quiet condition (*r* = 0.67)	78% (Excellent)
23	Koka et al. [29]	Electrocochleography in cochlear implant recipients with residual hearing: comparison with audiometric thresholds	2017	USA	20 Adults	Postoperative	Intracochlear	CM and ANN thresholds were strongly correlated with postop audiometric thresholds (CM: r^2^ = 0.87, ANN: r^2^ = 0.82).	78% (Excellent)
24	Koka et al. [24]	Intracochlear electrocochleography during cochlear implant electrode insertion is predictive of final scalar location	2018	USA	32 Adults	Intraoperative	Intracochlear	ECoG algorithm correctly estimated electrode position in 26 subjects, while 6 electrodes were wrongly identified as translocated.	78% (Excellent)
25	Mandala et al. [46]	Electrocochleography during cochlear implantation for hearing preservation	2012	Italy	27 Adults	Intraoperative	Round Window	Hearing preservation (<10 dB) at postop 1 m achieved in 11 out of 13 subjects with ECoG feedback.	78% (Excellent)
26	O’Connell et al. [23]	Intra- and postoperative electrocochleography may be predictive of final electrode position and postoperative hearing preservation	2017	USA	18 Adults	Intraoperative Postoperative	Intracochlear	No correlation found between intraop ECoG and postop audiometric thresholds. Postop threshold elevation was greater for electrodes with scalar dislocation.	89% (Excellent)
27	O’Leary et al. [47]	Intraoperative observational real-time electrocochleography as a predictor of hearing loss after cochlear implantation	2020	Australia	109 Adults	Intraoperative	Intracochlear	A total of 66 subjects with an ECoG drop during implantation had significantly poor hearing preservation at postop 3 m.	67% (Good)
28	Ramos-Macias et al. [48]	Intraoperative intracochlear electrocochleography and residual hearing preservation outcomes when using two types of slim electrode arrays in cochlear implantation	2019	Spain Australia	15 Adults	Intraoperative	Intracochlear	Subjects with overall increase in CM amplitude during insertion showed residual hearing (<15 dB). Subjects with a peak amplitude at the beginning and decrease afterwards had dropped hearing (15–30 dB).	89% (Excellent)
29	Riggs et al. [25]	Intracochlear electrocochleography: influence of scalar position of the cochlear implant electrode on postinsertion results	2019	USA	21 Adults	Intraoperative	Intracochlear	Translocation group showed 92% of a mean loss of preoperative pure tone average (PTA) compare to 52% in nontranslocation group.	89% (Excellent)
30	Scott et al. [49]	The compound action potential in subjects receiving a cochlear implant	2016	USA	130 Children 112 Adults	Intraoperative	Round Window	Weak correlation between CAP amplitudes and CNC scores (r^2^ = 0.25). ECoG TR accounted for 43% of the variance of CI outcomes.	78% (Excellent)

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
