# Peer review of "Electrocochleography in Cochlear Implant Users with Residual Acoustic Hearing: A Systematic Review"

_ijerph, 2020, doi:10.3390/ijerph17197043_

Round 1

Reviewer 1 Report

This review article describes the application of ECoG measurements in the clinical population of cochlear implant users with residual acoustic hearing. It is written well, organised clearly and concise. I believe this constitutes a valuable contribution to the scientific community and provides a helpful overview over the numerous articles that have been published on this matter. There is scientific justification due to the sheer number of studies that were deemed qualified for this review (30) and it is timely as the CI candidacy criteria have been widened in many countries in recent years to include people with lower levels of hearing loss (and more preserved acoustic hearing). The clinical goal to use ECoG for preserving hearing function is of strong interest in the community. The paper also does a good job in summarizing the main findings in those studies in a compressed style.

I only had minor comments, with the main one being: I am not sure calling it a “new” CI population is the best way to described the wider qualification criteria applied for CI candidacy. I suggest describing the population more precisely (“…with preserved acoustic hearing”) rather than calling them “new” (it may not be new anymore by the time this study is being read over the next 10 years).

Further minor corrections:

L11: “enjoy” may be a bit over the top, suggest changing to “with preserved hearing”

L34: conventional amplification “via hearing aids.”

L34: “appears”

L36: “techniques”

L45: delete “that” after periphery

L183: “A recent innovative technique…”

L188: “intracochlear”

L197: change to “nor” as you started with “Neither”

L201: This sentence is missing a verb.

L218: “adults”

L218: accounted “for” 46%

L227: may “be” clinically, “behavioral”, “populations”

L236: with “a” Cochlear

L237: users “were” presented

L242: delete “in”

L243: delete “also”

L245: status of “the” peripheral

L248: of “the” auditory

Table 1: most right column – just cosmetics: try to get “excellent” into one line?

Author Response

September 18, 2020

[IJERPH] Manuscript ID: ijerph-937035 - Major Revisions

Dear the editor and reviewers,

Thank you for the opportunity to revise my manuscript, Electrocochleography in Cochlear Implant Users with Residual Acoustic Hearing: A Systematic Review. I appreciate the careful review and constructive suggestions. I believe that the manuscript is substantially improved after making the suggested edits.

Following this letter are the reviewer comments with responses, including how and where the text was modified. Changes made in the manuscript and response letter below are marked in Red. This author has given approval to the final version of this revision.

Thank you for your consideration.

Sincerely,

Jeong-Seo Kim, Au.D.

# Reviewer 1

This review article describes the application of ECoG measurements in the clinical population of cochlear implant users with residual acoustic hearing. It is written well, organised clearly and concise. I believe this constitutes a valuable contribution to the scientific community and provides a helpful overview over the numerous articles that have been published on this matter. There is scientific justification due to the sheer number of studies that were deemed qualified for this review (30) and it is timely as the CI candidacy criteria have been widened in many countries in recent years to include people with lower levels of hearing loss (and more preserved acoustic hearing). The clinical goal to use ECoG for preserving hearing function is of strong interest in the community. The paper also does a good job in summarizing the main findings in those studies in a compressed style.

I only had minor comments, with the main one being: I am not sure calling it a “new” CI population is the best way to described the wider qualification criteria applied for CI candidacy. I suggest describing the population more precisely (“…with preserved acoustic hearing”) rather than calling them “new” (it may not be new anymore by the time this study is being read over the next 10 years).

: Thank you for your comments. As you suggested, I changed “a new CI population” to “this CI population with residual (or preserved) acoustic hearing” (please see lines 25 and 258).

Further minor corrections:

L11: “enjoy” may be a bit over the top, suggest changing to “with preserved hearing”

: Thank you for your suggestion. Revised as suggested.

L34: conventional amplification “via hearing aids.”

: Revised as suggested (please see line 33).

L34: “appears”

: Revised as suggested (please see line 34).

L36: “techniques”

: Revised as suggested (please see line 36).

L45: delete “that” after periphery

: Revised as suggested (please see line 45).

L183: “A recent innovative technique…”

: Revised as suggested (please see line 183).

L188: “intracochlear”

: Revised as suggested (please see line 188).

L197: change to “nor” as you started with “Neither”

: Revised as suggested (please see line 197).

L201: This sentence is missing a verb.

: Thank you for your comment. I added a verb (“were reported”) at the end of the sentence (please see lines 201-203).

L218: “adults”

: Revised as suggested (please see line 218).

L218: accounted “for” 46%

: Revised as suggested (please see line 218).

L227: may “be” clinically, “behavioral”, “populations”

: Revised as suggested (please see lines 227-228).

L236: with “a” Cochlear

: Revised as suggested (please see line 237).

L237: users “were” presented

: Revised as suggested (please see line 239).

L242: delete “in”

: Revised as suggested (please see line 243).

L243: delete “also”

: Revised as suggested (please see line 244).

L245: status of “the” peripheral

: Revised as suggested (please see line 246).

L248: of “the” auditory

: Revised as suggested (please see line 249).

Table 1: most right column – just cosmetics: try to get “excellent” into one line?

: Thank you for your suggestion. I adjusted a size of columns and made “Excellent” in the last column into one line.

Reviewer 2 Report

This review collates and summarizes the results of 30 published studies of ECoG measurements in cochlear-implant (CI) users with residual low-frequency hearing. This is a topic of some interest to the community of clinicians and hearing scientists who work with CI users; the relevance and timeliness of the subject is a strength of the manuscript. The manuscript does not attempt to use the dataset to answer a specific question. As a result, what is provided is a rather dry listing of results, leading to no particular conclusion. That is a weakness. The manuscript would have benefited from a careful proofreading; examples of text that could be improved are included in the specific comments below.

Specific comments:

line 10: ungrammatical sentence. possible change: "as a tool [for] assessing the response of the peripheral auditory system and monitor[ing] hearing preservation"

line 31: should be "criteria [have] been ..."

line 34: should be "appear[s] to be ..."

line 45: extra word "periphery [that] may ..."

line 89: one could say "was guided by" or "followed", but it's awkward to use both words together

line 95: should be "cochlea[r] implant"

lines 99-100: I'm not sure what "along with personal communication" means; please clarify

lines 153-154: There was no indication in the Methods that journal quality was evaluated or included as a search term of screening criterion. Either delete the statement about "top scientific journals", or provide an objective basis for the claim.

Author Response

September 18, 2020

[IJERPH] Manuscript ID: ijerph-937035 - Major Revisions

Dear the editor and reviewers,

Thank you for the opportunity to revise my manuscript, Electrocochleography in Cochlear Implant Users with Residual Acoustic Hearing: A Systematic Review. I appreciate the careful review and constructive suggestions. I believe that the manuscript is substantially improved after making the suggested edits.

Following this letter are the reviewer comments with responses, including how and where the text was modified. Changes made in the manuscript and response letter below are marked in Red. This author has given approval to the final version of this revision.

Thank you for your consideration.

Sincerely,

Jeong-Seo Kim, Au.D.

# Reviewer 2

This review collates and summarizes the results of 30 published studies of ECoG measurements in cochlear-implant (CI) users with residual low-frequency hearing. This is a topic of some interest to the community of clinicians and hearing scientists who work with CI users; the relevance and timeliness of the subject is a strength of the manuscript. The manuscript does not attempt to use the dataset to answer a specific question. As a result, what is provided is a rather dry listing of results, leading to no particular conclusion. That is a weakness. The manuscript would have benefited from a careful proofreading; examples of text that could be improved are included in the specific comments below.

: Thank you for your comments. I revised the manuscript as you suggested below.

Specific comments:

line 10: ungrammatical sentence. possible change: "as a tool [for] assessing the response of the peripheral auditory system and monitor[ing] hearing preservation"

: Thank you for your suggestion. Revised as suggested (please see lines 9 - 10).

line 31: should be "criteria [have] been ..."

: Revised as suggested (please see line 31).

line 34: should be "appear[s] to be ..."

: Revised as suggested (please see line 34).

line 45: extra word "periphery [that] may ..."

: Revised as suggested (please see line 45).

line 89: one could say "was guided by" or "followed", but it's awkward to use both words together

: Thank you for your comment. I changed it to “was guided by” and delete “following” (please see line 89).

line 95: should be "cochlea[r] implant"

: Revised as suggested (please see line 95).

lines 99-100: I'm not sure what "along with personal communication" means; please clarify

: Thank you for your comment. I changed the sentence to “The reference lists of retrieved articles and personal communication were also used to search potentially relevant articles.” (Please see lines 98 – 100).

lines 153-154: There was no indication in the Methods that journal quality was evaluated or included as a search term of screening criterion. Either delete the statement about "top scientific journals", or provide an objective basis for the claim.

: Thank you for your comment. As you suggested, I deleted the statement about “top scientific journals” and changed the sentence to “This indicates that included studies are mostly high quality.”

Round 2

Reviewer 2 Report

The author has been very responsive to suggestions.  My comments were adequately addressed in the revised version.